# Non-coding *cis*-element of *Period2* is essential for maintaining organismal circadian behaviour and body temperature rhythmicity

Masao Doi [1,7,8], Hiroyuki Shimatani[1,7], Yuta Atobe[1,7], Iori Murai[1,2], Hida Hayashi[1], Yukari Takahashi[1], Jean-Michel Fustin [1], Yoshiaki Yamaguchi[1], Hiroshi Kiyonari[3], Nobuya Koike[4], Kazuhiro Yagita[4], Choogon Lee[5], Manabu Abe[6], Kenji Sakimura[6] & Hitoshi Okamura[1,2,8]

Non-coding *cis*-regulatory elements are essential determinants of development, but their exact impacts on behavior and physiology in adults remain elusive. *Cis*-element-based transcriptional regulation is believed to be crucial for generating circadian rhythms in behavior and physiology. However, genetic evidence supporting this model is based on mutations in the protein-coding sequences of clock genes. Here, we report generation of mutant mice carrying a mutation only at the E′-box *cis*-element in the promoter region of the core clock gene *Per2*. The *Per2* E′-box mutation abolishes sustainable molecular clock oscillations and renders circadian locomotor activity and body temperature rhythms unstable. Without the E′-box, *Per2* messenger RNA and protein expression remain at mid-to-high levels. Our work delineates the *Per2* E′-box as a critical nodal element for keeping sustainable cell-autonomous circadian oscillation and reveals the extent of the impact of the non-coding *cis*-element in daily maintenance of animal locomotor activity and body temperature rhythmicity.

[1] Department of Systems Biology, Graduate School of Pharmaceutical Sciences, Kyoto University, Sakyō-ku, Kyoto 606-8501, Japan. [2] Laboratory of Molecular Brain Science, Graduate School of Pharmaceutical Sciences, Kyoto University, Sakyō-ku, Kyoto 606-8501, Japan. [3] Laboratories for Animal Resource Development and Genetic Engineering, RIKEN Center for Biosystems Dynamics Research, Kobe 650-0047, Japan. [4] Department of Physiology and Systems Bioscience, Kyoto Prefectural University of Medicine, Kyoto 602-8566, Japan. [5] Department of Biomedical Sciences, College of Medicine, Florida State University, Tallahassee FL 32306, USA. [6] Department of Cellular Neurobiology, Brain Research Institute, Niigata University, Niigata 951-8585, Japan. [7] These authors contributed equally: Masao Doi, Hiroyuki Shimatani, Yuta Atobe. [8] These authors jointly supervised this work: Masao Doi, Hitoshi Okamura. Correspondence and requests for materials should be addressed to M.D. (email: doimasao@pharm.kyoto-u.ac.jp) or to H.O. (email: okamurah@pharm.kyoto-u.ac.jp)

Evidence shows that non-coding *cis*-regulatory elements are as important as protein-coding sequences for determining cell identity and morphological development[1–3]. The degree of the effects of non-coding *cis*-regulatory elements on animal behavior and physiology after development, however, remains elusive. Circadian clocks generate ~24 h rhythms in behavior and physiology, which allow organisms to anticipate and adjust to daily environmental changes. The rhythm generating mechanism of the circadian clock involves clock genes, which regulate their own transcription in a negative transcription-translation feedback loop[4–6]. In this canonical feedback model, non-coding *cis*-elements are fundamental to both initiating and closing the loop[7–9]. Genetic evidence supporting this model, however, is still exclusively based on the effects of mutations in the protein-coding sequences of the clock genes. Therefore, while it is tempting to speculate that the circadian clock-related non-coding *cis*-elements are operational for daily dynamic regulation of behavior and physiology, there is currently no direct evidence to corroborate this notion.

Notwithstanding the recent intriguing discoveries of circadian oscillations in peroxiredoxin superoxidation in transcriptionally incompetent anucleate erythrocytes[10] and the expression of such cycles in the neurons in the suprachiasmatic nucleus (SCN)[11], the consensus conjecture regarding the mammalian clockwork supports the transcriptional feedback model. However, the literature addressing mutations or deletions in the protein-coding sequences of the clock genes cannot negate the possible contribution of post-translational (or non-genomic) clock mechanisms. Moreover, the extent of the impact of the *cis*-regulated transcriptional feedback cycle on establishing circadian rhythmicity in vivo remains the subject of considerable debate[4,12], due in large part to the lack of genetic research on non-coding *cis*-element mutations.

In the current study, we develop mutant mice carrying a mutation in a circadian *cis*-acting element of the core clock gene *Per2*. Although *Per2* was cloned as a secondary *period* gene in mammals, gene knockout studies revealed that *Per2* mutant mice displayed a loss of circadian rhythmicity, revealing its prominent role in the mammalian molecular clockwork[13–15]. Moreover, familial advanced sleep phase syndrome in humans is attributed to a missense mutation in the *Per2* gene[16].

The *Per2* E′-box sequence (5′–CACGTT–3′) located near the putative transcription initiation site[17] (–20 to –15) has been demonstrated to be the principal circadian *cis*-element that is sufficient to induce oscillating levels of reporter transcription via the mouse *Per2* minimal promoter[17,18]. The importance of this particular *cis*-element is further implied by the high degree of DNA sequence conservation in its flanking region between humans and mice[17] and by extremely enriched clock protein binding to this element[19]. Publicly available genome-wide ChIP-seq data highlight the predominant peak of clock protein binding activity around this E′-box (see Supplementary Fig. 1).

The present study was designed to investigate the role of this unique E′-box sequence of the *Per2* promoter (hereafter, *Per2* E′-box) as a potential nodal *cis*-element in the mammalian clockwork. By introducing a site-specific mutation at this element in vivo, we provide empirical evidence to show that the *Per2* E′-box is essential for maintaining cell-autonomous circadian oscillations. The cells without the *Per2* E′-box cannot maintain circadian molecular oscillations in culture conditions. At the organismal level, mice lacking the *Per2* promoter E′-box show destabilized locomotor activity and body temperature rhythms under altered light conditions, including constant light (LL) and experimental jet-lag conditions. Due to compensatory mechanisms in vivo, the mutant mice kept under constant dark (DD) conditions remain rhythmic but exhibit considerably shorter circadian periods than WT mice. Our data therefore define the

degree of the impact of the deletion of the *Per2* E′-box on the organismal clock: The *Per2* E′-box is essential for the period determination of behavioral rhythms in DD conditions and for sustaining stable rhythms under LL and jet-lag conditions. These data underscore the roles of the non-coding *cis*-element in the regulation of daily behavior and physiology in adulthood.

## Results

**Generation of *Per2* E′-box mutant mice**. We developed mutant mice harboring a targeted mutation in only the *Per2* E′-box. To do this, we used the *piggyBac* (*PB*) transposase tool for genome engineering because it allows for seamless removal of the *PB*-flanked marker cassette (*PB*-Neo) from the host genome after the mutation is introduced (Fig. 1a). Conventional methods that rely on the Cre/*lox*P or Flp/*FRT* system leave behind a single *lox*P- or *FRT*-derived ectopic sequence after marker excision. Such a footprint sequence could affect promoter architecture and/or enhancer communication of the target gene[20]. The PB transposase (PBase)/*PB* system circumvents this problem[21] (see Fig. 1b–d). Southern blot analysis confirmed that the *PB*-neomycin cassette was deleted without re-integration into the host genome (Fig. 1c). The footprint-free disappearance of the marker cassette was further confirmed by DNA sequencing (Fig. 1d). To reduce the confounding effects of a mixed genetic background, the mutant mice were backcrossed with C57BL/6J mice until microsatellite markers covering all individual chromosomes were congenic to the C57BL/6J strain (see Supplementary Fig. 2). The targeted mutation of the E′-box sequence was finally verified by DNA sequencing using homozygous mutant mice (Fig. 1d). Chromatin immunoprecipitation (ChIP) assays using homozygous mutant mouse liver, sampled at 4 h intervals for 24 h under DD conditions, revealed that the clock proteins PER1, CRY1, CRY2, and CLOCK had consistently low binding levels to the mutated *Per2* promoter (Fig. 1e). In a control experiment with WT (+/+) samples, the peak binding levels of PER1, CRY1, CRY2, and CLOCK to the *Per2* promoter were observed at CT12, CT4, CT20, and CT8, respectively, consistent with previous reports[19,22]. These results confirmed that *Per2* E′-box function was abolished in the mutant mice.

***Per2* E′-box is essential to maintain cellular circadian oscillations**. To investigate the role of the *Per2* E′-box in maintaining cellular oscillation, we generated primary fibroblast cultures from WT (*Per2*E′⁺/⁺) and mutant (*Per2*E′ᵐ/ᵐ) mouse lung tissues and monitored circadian fluctuations in the PER2 protein levels over 80 h (Fig. 2a). As expected, the endogenous PER2 protein in synchronized WT cells displayed characteristic circadian oscillations in both abundance and electrophoretic mobility, which continued over multiple cycles under constant culture conditions[23] (Fig. 2a). In contrast, *Per2*E′ᵐ/ᵐ cells failed to maintain normal PER2 protein oscillations (Fig. 2a). It is important to note that the mutant cells could competently form the first surge of PER2 expression after synchronization (4–12 h). However, following a subsequent increase (28 h), PER2 expression in the mutant cells remained at mid-to-high levels and eventually lost apparent circadian variation after 60 h (see also Supplementary Fig. 3). Electrophoretic migration rhythms also disappeared from the mutant cells (Fig. 2a). The relative intensities of the three major bands of the PER2 protein remained unchanged in the *Per2*E′ᵐ/ᵐ cells. The post-translational rhythms were thus affected either directly or indirectly by the mutation of the E′-box. We experimentally confirmed that the protein-coding sequences of PER2 and known clock protein kinases and phosphatases were unaffected in our mutant mice. These results indicate that the E′-box is indispensable for maintaining normal PER2 protein oscillations.

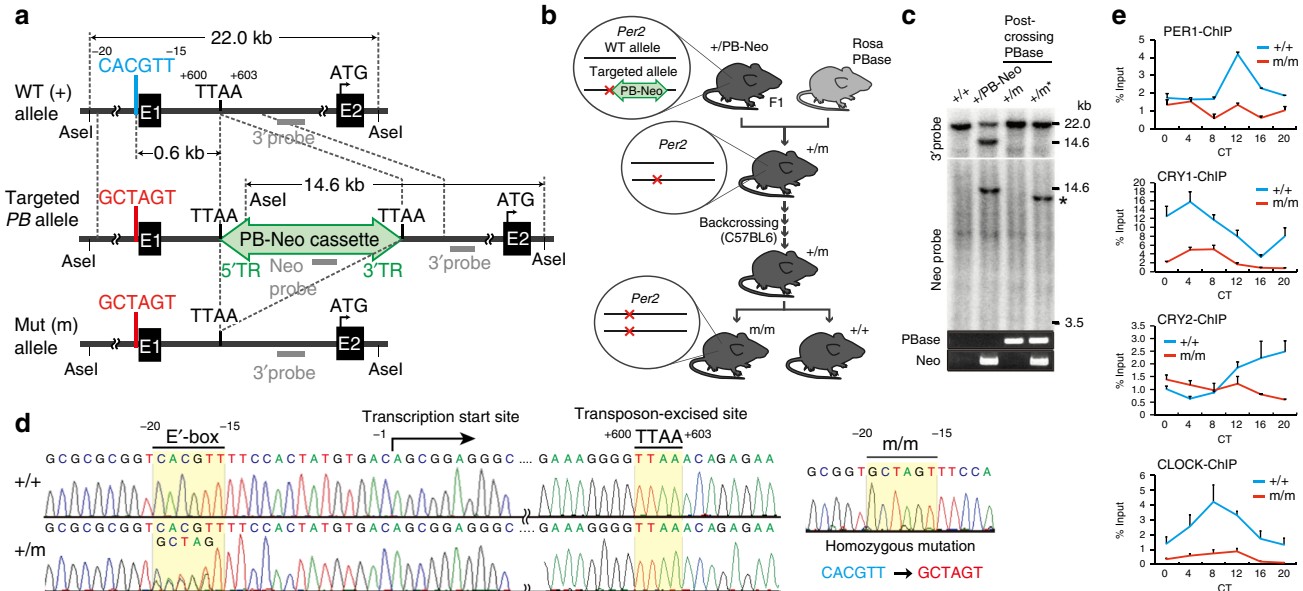

**Fig. 1** Generation of mice carrying a targeted mutation at *Per2* E′-box. **a** Genome modification strategy using the *piggyBac* transposon. The CACGTT *Per2* E′-box was mutated to GCTAGT. Top line, genome architecture of the mouse *Per2* gene; E1, exon 1; E2, exon 2; Blue, wildtype E′-box; Red, mutated E′-box; Green, *piggyBac* transposon carrying a neomycin-resistant gene (PB-Neo cassette) inserted at a genomic TTAA site (+600 to +603); Gray bars, Southern blot probes. Numbering shows the position relative to the putative transcription start site (+1). **b** Interbreeding scheme. Mice heterozygous for the targeted allele (+/PB-Neo) were intercrossed with ROSA26-PBase mice. The resultant mutant mice without the marker cassette (+/m) were backcrossed into the C57BL/6J strain. **c** Southern blot and PCR analysis showing the insertion (+/PB-Neo) and excision (+/m and +/m*) of the *piggyBac* transposon. *, a reintegrated PB-Neo fragment. **d** Genomic DNA sequences of WT (+/+) and *Per2* E′-box mutant (+/m and m/m) mice, illustrating the precise mutation of the E′-box and seamless excision of the PB-Neo cassette from the TTAA site. Heterozygous (+/m) mouse sequences exhibit dual signals for CACGTT and GCTAGT at the E′-box. **e** ChIP values for WT (+/+) and homozygous mutant (m/m) mouse liver sampled at 4 h intervals for 24 h on the first day in DD. Values are means ± s.e.m. of three technical replicates. Source data for (**c**) and (**e**) are provided as a Source Data file

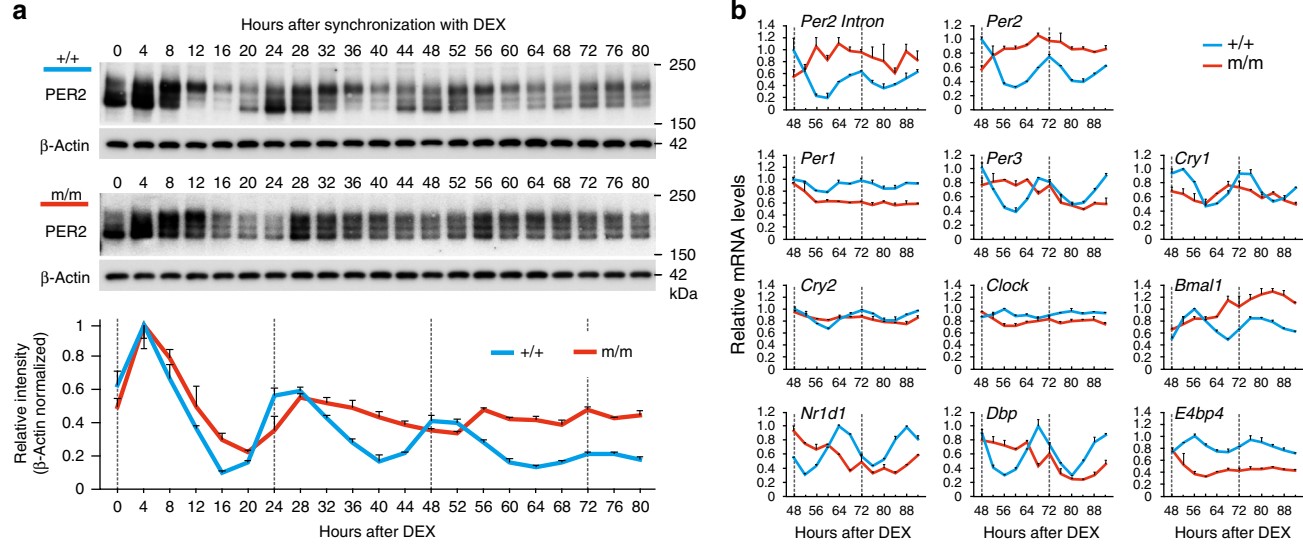

**Fig. 2** *Per2* E′-box is essential for maintaining cell-autonomous circadian oscillations. **a** Temporal profiles of PER2 protein expression in *Per2*E′+/+ and *Per2*E′m/m fibroblasts. Representative immunoblots and normalized densitometry values (*n* = 3, mean ± s.e.m.) are shown. **b** mRNA profiling of clock genes in *Per2*E′+/+ and *Per2*E′m/m fibroblasts (*n* = 2, for each data point). For *Per2*, both intron and exon RNA were analyzed. The data are presented as the mean ± variation. Source data are provided as a Source Data file

In the *Per2*E′m/m cells, we observed that both *Per2* mRNA and pre-mRNA (intronic RNA) levels remained constitutively high, with values exceeding those of the circadian peak in the WT cells at 72 h (Fig. 2b). Thus, *Per2* transcription remains active even without this particular E′-box sequence in the promoter. Moreover, extensive mRNA profiling revealed that the effects of the mutation of the *Per2* E′-box were not limited

to *Per2* transcription. The dysfunctional *Per2* E′-box also abolished the circadian expression of other clock genes and output genes, including *Per1*, *Per3*, *Cry1*, *Cry2*, *Bmal1*, *Nr1d1*, *Dbp*, and *E4bp4* (Fig. 2b). These pervasive effects of the mutation provide evidence that the *Per2* E′-box is a fundamental *cis*-element that maintains normal molecular clock oscillations.

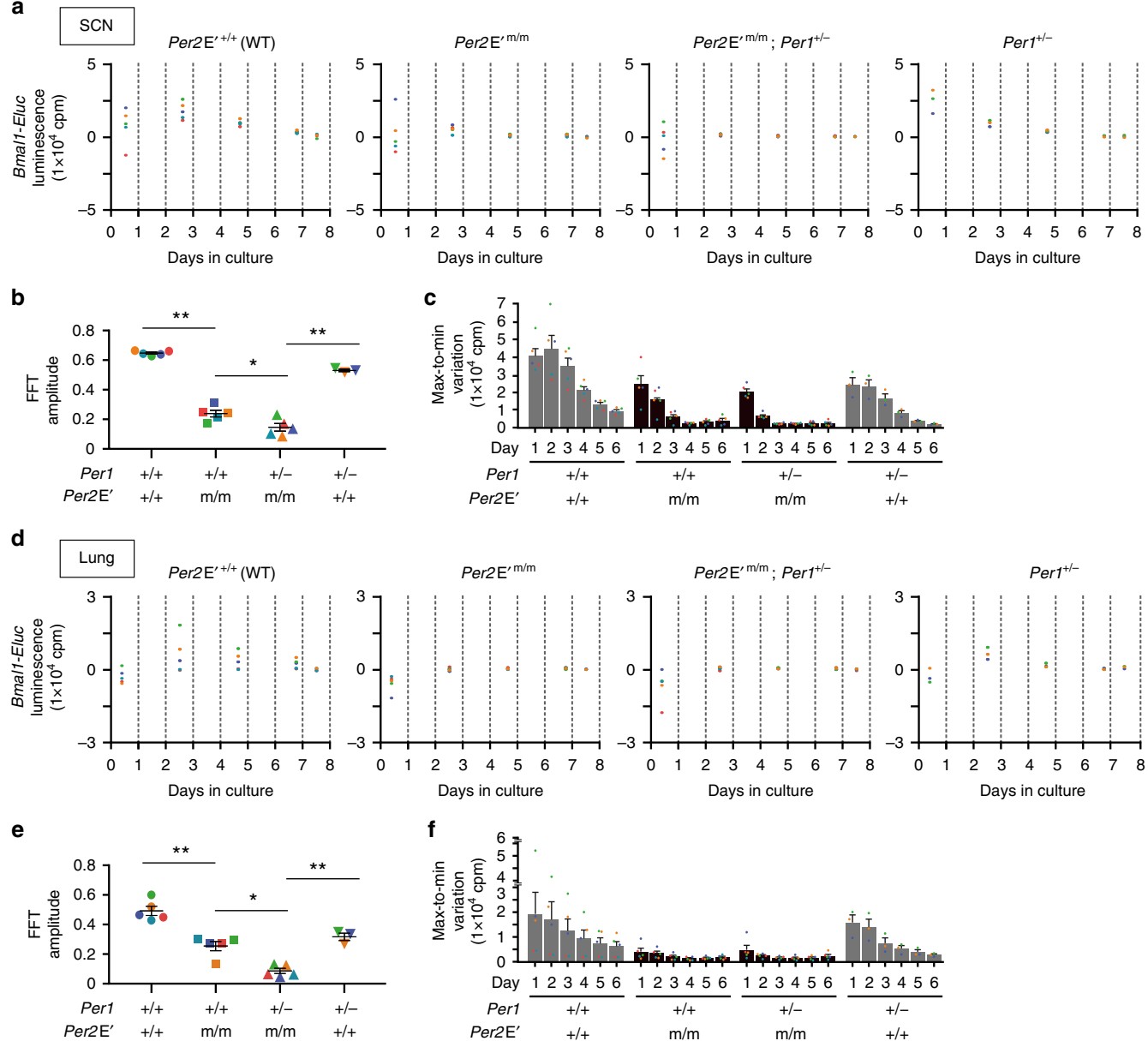

**Fig. 3** *Per2* E′-box is essential for sustainable *Bmal1* oscillations in SCN and lung explants. **a** *Bmal1-Eluc* bioluminescence traces of ex vivo SCN cultures from *Per2*E′⁺/⁺ (n = 5), *Per2*E′m/m (n = 5), *Per2*E′m/m; *Per1*⁺/⁻ (n = 5), and *Per1*⁺/⁻ (n = 3) mice. Averaged de-trended data are shown. **b** FFT amplitude of (**a**). *P < 0.05, **P < 0.001, one-way ANOVA, Bonferroni post hoc test. **c** Daily max-to-min variations of (**a**). The data are presented as the mean ± s.e.m. **d** *Bmal1-Eluc* bioluminescence traces of lung tissue explant cultures from *Per2*E′⁺/⁺ (n = 5), *Per2*E′m/m (n = 5), *Per2*E′m/m; *Per1*⁺/⁻ (n = 5), and *Per1*⁺/⁻ (n = 3) mice. Averaged de-trended data are shown. **e** FFT amplitude of (**d**). *P < 0.05, **P < 0.001, one-way ANOVA, Bonferroni post hoc test. **f** Daily max-to-min variations of (**d**). The data are presented as the mean ± s.e.m. Source data for (**c**) and (**f**) are provided as a Source Data file

***Per2* E′-box is essential to maintain molecular oscillations in the SCN.** The SCN in the hypothalamus is the primary regulator of daily rhythms of behavior and physiology in mammals[24,25]. We next examined the effect of the *Per2* E′-box mutation on the SCN clock. To study the molecular rhythms in the SCN, we used a *Bmal1* promoter-driven luciferase reporter (*Bmal1-Eluc*)[26], as it allowed the assessment of molecular rhythms that are not a simple reflection of the *Per2* loop. In agreement with previous reports[27], organotypic SCN slices prepared from control mice (*Per2*E′⁺/⁺; *Bmal1-Eluc*) displayed persistent circadian rhythms of luminescence, which continued for over a week in culture (Fig. 3a, see also Supplementary Fig. 4). In contrast, all tested SCN slices from the *Per2* E′-box mutant mice (*Per2*E′m/m;*Bmal1-Eluc*) displayed attenuated rhythms of luminescence, which were

damped within 2–3 cycles (Fig. 3a). Fast Fourier transform (FFT) analysis of the de-trended waveforms of days 1.5−7.5 (Fig. 3b) confirmed the reduced rhythmicity in the *Per2*E′m/m SCN. This finding indicated the relevance of the *Per2* E′-box in maintaining normal molecular circadian oscillations in the SCN. Nevertheless, detectable rhythms were noted in the attenuated *Per2*E′m/m SCN slices over the initial 2–3 cycles (Fig. 3a). These remaining rhythms were likely dependent on *Per1*, because SCN explants from *Per2*E′m/m; *Per1* heterozygous null mice (*Per2*E′m/m; *Per1*⁺/⁻;*Bmal1-Eluc*) exhibited even fewer persistent rhythms, which were damped within 2 cycles (Fig. 3a–c). Importantly, similar attenuation of *Bmal1-Eluc* rhythmicity by *Per2*E′m/m and further by *Per1*⁺/⁻ could be reproduced in cultures of lung (Fig. 3d–f) and adrenal explants (Supplementary Fig. 4),

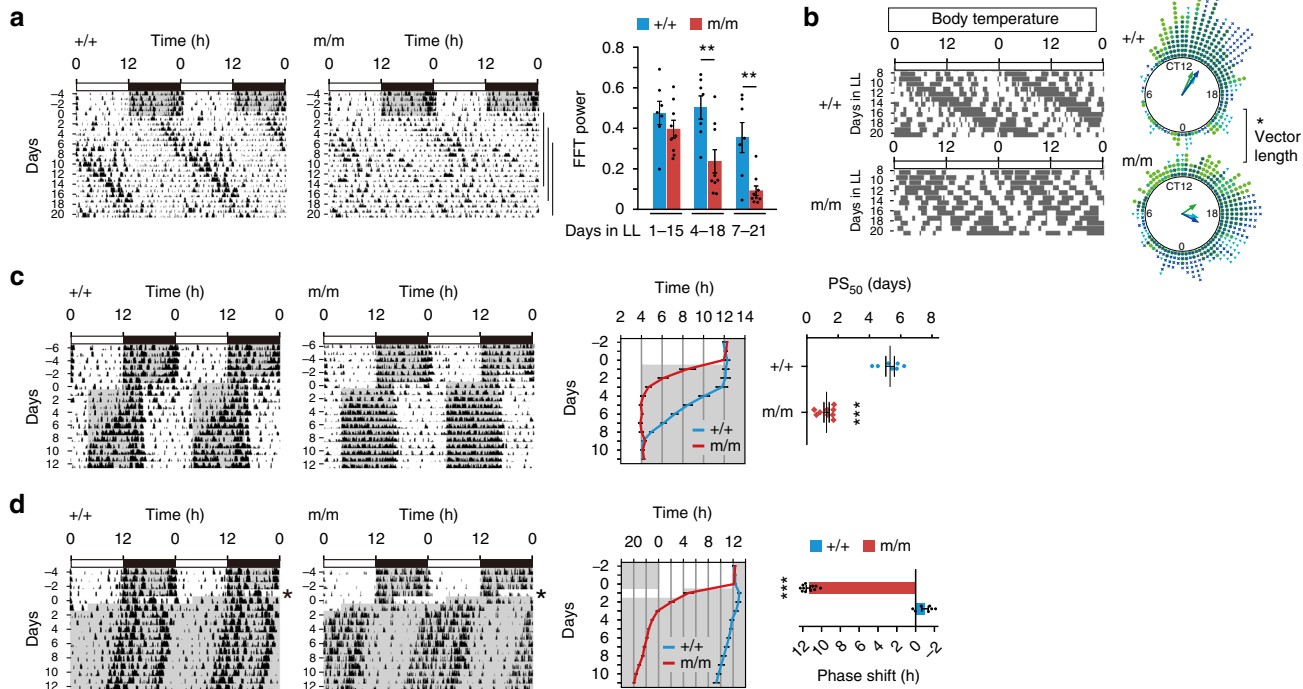

**Fig. 4 a** Destabilized circadian locomotor activity and body temperature rhythms of *Per2* E′-box mutant mice. Representative locomotor activity records of *Per2*E′[+/+] and *Per2*E′[m/m] mice under light/dark (LD) followed by constant light (LL). The graph shows changes in FFT power in LL. **$P < 0.01$, two-way repeated-measures ANOVA, Bonferroni post hoc test (*Per2*E′[+/+], $n = 7$; *Per2*E′[m/m], $n = 9$). **b** Representative body temperature records of *Per2*E′[+/+] and *Per2*E′[m/m] mice in LL. The mean temperature of the entire data series was calculated, and the data points above the mean are plotted. Rayleigh plots show phase distribution of elevated body temperature on days 7–21 ($n = 3$, each genotype). Data from independent animals are color-coded. Arrow length reflects the *r*-value of each distribution. *$P < 0.05$, two-tailed unpaired *t*-test. **c** Representative records of locomotor activity (left) and plots of activity onset (middle) of *Per2*E′[+/+] and *Per2*E′[m/m] mice before and after an 8-h phase advance in LD cycles. The PS$_{50}$ values represent the time required for 50% phase-shift (right). ***$P < 0.0001$, two-tailed unpaired t-test (*Per2*E′[+/+], $n = 7$; *Per2*E′[m/m], $n = 9$). **d** Representative locomotor activity records (left), activity onset (middle), and magnitude of phase-shift (right) of *Per2*E′[+/+] and *Per2*E′[m/m] mice subjected to an 8-h phase advance on day 1 and released to DD. ***$P < 0.0001$, two-tailed unpaired *t*-test (*Per2*E′[+/+], $n = 7$; *Per2*E′[m/m], $n = 11$). The data are the mean ± s.e.m. Source data are provided as a Source Data file

confirming the importance of the *Per2* E′-box in keeping sustained molecular circadian oscillations not only for the SCN but also for the peripheral tissues.

**Destabilized organismal rhythms of animal lacking *Per2* E′-box.** Finally, to assess the effects of the mutation at the behavioral level, C57BL/6J-backcrossed WT and *Per2*E′[m/m] mice were housed in a 12-h light:12-h dark (LD) cycle and then subjected to DD, LL, or experimental jet-lag conditions. Actograms of the animals in the respective conditions were acquired (see Supplementary Figs. 5–9). Interestingly, mutant mice kept in DD showed overt circadian locomotor activity rhythms, although with a significantly shortened free-running circadian period relative to WT mice (WT vs. *Per2*E′[m/m]: 23.71 ± 0.02 h vs. 23.50 ± 0.02 h; see Supplementary Fig. 5). This contrast with the in vitro data suggests that a compensatory mechanism functions in vivo. We thus examined the circadian behavior of *Per1*[−/−]; *Per2*E′[m/m] double-deficient mice (see Supplementary Fig. 6). Although their period length was unstable and changed gradually over extended exposure to constant darkness, *Per1*[−/−]; *Per2*E′[m/m] mice displayed behavioral rhythms over 40 days. With analogy, *Drosophila* strains expressing *period* or *timeless* under a constitutive promoter were reported to show behavioral rhythms with an altered period length[12,28]. These data suggest that organisms might have a compensatory mechanism to mitigate defective transcriptional feedback regulation[4,24,29].

Compared to the modest phenotype in DD, we observed unstable rhythmicity of the mutant mice in LL (Fig. 4a). When placed in LL for 3 weeks, WT subjects ($n = 7$) stayed rhythmic with a period longer than 24 h, which is consistent with data from previous reports[30,31]. However, none of the *Per2*E′[m/m] mice ($n = 9$) remained rhythmic (Fig. 4a, see also Supplementary Fig. 7); all tested mutant mice displayed behaviors characterized by a gradual decrease in the power of rhythmicity (FFT spectrogram, Fig. 4a). Since light has a negative masking effect on locomotor activity, we measured core body temperature fluctuations and confirmed that the body temperature recapitulated the reduced rhythmicity of the mutant mice in LL conditions (Fig. 4b, Rayleigh vector length of WT vs. *Per2*E′[m/m]: 0.64 ± 0.06 vs. 0.42 ± 0.02).

The *Per2* E′-box mutant mice were also distinct under experimental jet-lag conditions (Fig. 4c, see also Supplementary Fig. 8). When the ambient LD cycle was advanced by 8 h, WT mice re-entrained progressively over 8 to 9 days[32]. In contrast, the *Per2*E′[m/m] mice adapted to the new cycle within 2–3 days. The 50% phase-shift value (PS$_{50}$)[32], measured from activity onset, indicated a rapidity of 5.33 ± 0.28 days for WT mice and 1.26 ± 0.17 days for *Per2*E′[m/m] mice (Fig. 4c). Notably, the difference between the two genotypes was even pronounced when the animals were exposed to a single 8-h light advance, followed by DD conditions (Fig. 4d, see also Supplementary Fig. 9). Under this light regime, all tested mutant mice shifted forward by approximately 12 h, which is nearly 180° out of phase in a 24-h

cycle, relative to WT mice (Fig. 4d). The underlying mechanism of this extremely large shift is unknown. While this could be due to an indirect effect of the weak clock, a limit cycle oscillator model[33] predicts that a reduced-amplitude pacemaker in the mutant mice (reduced radius of the limit cycle) could have this effect[34]. We also noticed that after a light pulse exposure, the mutant mice show a slightly enhanced *Per2* mRNA induction in the SCN (see Supplementary Fig. 10). This perhaps also partly contributes to the enhanced resetting of the mutant mice.

## Discussion

Non-coding *cis*-regulatory elements are known to play a critical role in development, but their precise effects on daily behavior and physiology in adulthood remain elusive. The present study was designed to provide genetic evidence for the roles of the circadian *cis*-element in vivo. Despite the presence of compensatory mechanisms in vivo, our work shows that the *Per2* E′-box is essential for maintaining optimal locomotor activity rhythms under LL conditions and enabling the phase of the clock to resist abrupt shifts in LD cycling. In general, adapting to new LD cycles is rarely instantaneous and requires repeated 24 h cycles. However, the *Per2* E′-box mutant mice adapted to a new phase immediately. These data elucidate the impact of the deletion of the non-coding *cis*-element in daily maintenance of behavioral activity in adulthood.

In addition, our cell culture data reinforce the concept that circadian clock phenotypes are more drastic at the cellular level than the organismic level[4,24]. *Per2* E′-box mutation leads to unsustainable gene expression rhythms in organotypic SCN slices and cultured peripheral tissues and fibroblasts. These observations substantiate the general conjecture that *cis*-regulatory element-based gene transcription is essential for sustaining cellular clock oscillations.

Not surprisingly, the overall behavioral phenotypes of the *Per2* E′-box mutant mice reveal that the *Per2* E′-box is not an absolute requirement for behavioral rhythm generation. Under DD conditions, its deletion affects only the circadian period (Supplementary Figs. 5 and 6), a phenotype analogous to that observed in transgenic *Drosophila* strains expressing *period* or *timeless* via a constitutive promoter[12,28]. At the gene expression level, *Per2* expression in the liver and SCN in the DD-kept mutant mice in vivo was still rhythmic, albeit with an increased baseline (see Supplementary Fig. 11). These observations suggest that organisms have a compensatory mechanism to mitigate defective transcriptional feedback regulation. Correspondingly, previous studies demonstrate that behavioral rhythms do not necessarily reflect cellular clock phenotypes[29,35–37]; superior circadian performance of behavioral rhythmicity has been observed in several clock gene mutant mice, compared to tissue cultures obtained from the same animal strains[29]. In vivo multicellular and/or inter-organ systemic circuitry might compensate for poor core clock function within individual cells[4]. In this regard, it is worth noting that systemic extracellular signals are known to affect the *Per2* promoter via non-E′-box *cis*-regulatory elements, such as $Ca^{2+}$/cAMP response element (CRE)[38] and glucocorticoid response element (GRE)[39]. Extracellular circadian feedback pathways through these non-E′-box elements might contribute to the compensation mechanisms in vivo[40,41].

Our work differs from transgenic studies in which non-native promoters are used[12,17]. Transgenic studies and rescue experiments can be affected by shorter promoter regulatory sequences and position effects. In comparison, native promoters are characterized by endogenous enhancer elements and normal chromatin structure, allowing for the preservation of regulation by epigenetic factors, which are known to be crucial for controlling transcription[5]. Under

these near-native conditions, we noticed that the deletion of the *Per2* E′-box leads to accumulation of *Per2* mRNA and protein in cultured fibroblasts. Constitutively un-suppressed transcription of *Per2* likely underlies the constant accumulation of PER2 protein and resultant compromised gene expression rhythms in the mutant cells.

*Cis*-element provides a place where both active and repressive transcription complexes are recruited. The net effect of its absence thus seems context-dependent. The question of this context specificity is still unresolved in the field of transcription and chromatin remodeling. Particularly, in our case, the mechanism(s) that allows the continued transcription of *Per2* without relying on the E′-box is still unknown. It is possible that native chromatin structure of the *Per2* might permit its transcription. It is also conceivable that other circadian *cis*-elements on the *Per2* promoter, such as D-box and CRE, might contribute to the basal transcription of *Per2*[9,18,42]. In this regard, the continued expression of *Per2* might reflect a confounding effect of being chronically deficient in the functional E′-box. Given that *Per2* is under regulation of interlocked feedback cycles[6,9], remaining *Per2* transcription might be a homeostatic consequence of a nonfunctional clock in the mutant cells. A complete understanding of circadian regulation of *Per2* in vivo under native conditions remains a challenge for future study.

Genome-wide association studies recently identified many non-coding variants that account for human chronotypes[43–45]. Given the physiologic relevance of the *Per2* cis-element, a targeted point mutation strategy would facilitate hypothesis-driven approaches to understand the extent of the impact of the non-coding elements on the daily physiology and pathophysiology of the organism.

## Methods

**Generation of the *Per2* E′-box mutant mice**. The CACGTT E′-box sequence located in the *Per2* promoter (–20 to –15; +1, the putative transcription start site[17]) was targeted. Note that numbering of the relative position of the E′-box sequence will change according to the definition of the transcription initiation site[17,18] (see also RefSeq database in the UCSC genome browser: https://genome.ucsc.edu/). CACGTT was mutated to GCTAGT, because this change is reported to abrogate CLOCK:BMAL1-mediated activation of *Per2* promoter activity in vitro[17]. We employed *piggyBac* (*PB*) transposon-based gene engineering system[46,47], because conventional methods relying on Cre/*lox*P or Flp/*FRT* systems leave behind a single *lox*P or *FRT*-derived sequence after excision, which causes the creation of 'footprint' (residual ectopic sequence) that might affect promoter architecture and/or enhancer communication of target gene[48]. The targeting vector to generate *PB* mutant allele of *Per2* was constructed by using a Red/ET recombination system (Gene Bridges). The bacterial artificial chromosome (BAC) containing *Per2* was obtained from BACPAC Resources at the Children's Hospital and Research Center at Oakland (RP23-343F13). With a Red/ET cloning method (Gene Bridges), a 10-kb genomic region of *Per2* (–5899 to +4,103; +1, the putative transcription start site[17]) that corresponds to the region of 6.5 kb upstream (long arm) and 3.5 kb downstream (short arm) of the *Per2* E′-box (–20/–15) was cloned into a vector containing a diphtheria toxin A gene (pDTA vector). Then, a DNA amplicon of *Per2* (–90 to +650) that contains a mutated E′-box (GCTAGT) at –20/–15 was obtained by fusion PCR, and this was further modified by inserting a *PB* terminal repeat sequence (5′TR and 3′TR)-flanked neomycin cassette (*PB*-NEO) into the TTAA quadruplet sequence of *Per2* (+600/+603). With a second-round Red/ET cloning, this fragment was recombined into the 10-kb *Per2* BAC/pDTA vector. The resultant *Per2* targeting vector was verified by DNA sequencing.

Gene targeting was carried out with TT2 embryonic stem cell. Germline transmission was verified by PCR and Southern blotting. The *Per2* E′-box mut *PB*-NEO F1 mice (Accession No. CDB1060K: http://www2.clst.riken.jp/arg/mutant%20mice%20list.html) were intercrossed with ROSA26-PBase knock-in mice[49] (a gift from A. Bradley, Wellcome Trust Sanger Institute, UK) to remove the *PB*-NEO cassette from the genome. A seamless excision of the cassette was verified by direct DNA sequencing. Genotypes of the E′-box were determined by DNA sequencing and/or TaqMan qPCR with the following probes: WT, 5′-FAM-TAG TGG AAA ACG TGA CCG C-MGB-3′, and mutant, 5′-VIC-TAG TGG AAA CTA GCA CCG C-MGB-3′. The use of different reporter dyes with separated emission wavelength maxima (FAM and VIC) enabled concurrent detection of the two alleles in a single PCR with a common primer set for *Per2*: forward, 5′-GGA GCC GCT AGT CCC AGT AG-3′; reverse, 5′-AGG TGG CAC TCC GAC CAA T-3′. The established mutant mice were backcrossed into the C57BL/6J background using a marker-assisted breeding (i.e., speed congenic) approach[50,51]. Where specified, *Per2*E′m/m mice were intercrossed with mice carrying a *Bmal1-Eluc* reporter[26,27,52] and/or *Per1*-deficient mice[15] of C57BL/6J background (The Jackson Laboratory, 10491).

All animal experiments were conducted in compliance with ethical regulations in Kyoto University and performed under protocols approved by the Animal Care and Experimentation Committee of Kyoto University and Institutional Animal Care and Use Committee of RIKEN Kobe Branch.

**ChIP**. ChIP assays were performed with technical replicates[19,53]: We repeated chromatin/DNA shearing with equal amounts of aliquots from the same liver nuclear sample. In brief, whole liver sample was homogenized with ice-cold PBS (4 ml/liver) and cross-linked in two steps using first 2 mM disuccinimidyl glutarate for 20 min then 1% methanol-free ultrapure formaldehyde for 5 min at room temperature. Glycine was added for 5 min in 125 mM final concentration. The homogenates (~5 ml/liver) were mixed with ice-cold 2.3 M sucrose buffer (10 ml/ liver) containing 125 mM glycine, 10 mM HEPES pH 7.6, 15 mM KCl, 2 mM EDTA, 0.15 mM spermine, 0.5 mM spermidine, 0.5 mM DTT, and 0.5 mM PMSF and layered on top of a 5 ml cushion of 1.85 M sucrose. Ultracentrifugation was performed at $105,000 \times g$ for 1 h at 4 °C with a Beckman SW28 rotor. The resultant nuclear pellets were stored at −80 °C until use. The nuclei were resuspended in 1.5 ml per liver of IP buffer (10 mM Tris-HCl pH 7.5, 150 mM NaCl, 1 mM EDTA, 1% Triton X-100, 0.1% sodium deoxycholate, 1 mM PMSF, protease inhibitor cocktail) and divided equally into three aliquots, which were each separately sonicated around 15 s for 80 times at 4 °C using a Bioruptor UCW-201TM apparatus (Tosho Denki, Japan). For each reaction, 10 μg fragmented chromatin was resuspended in 500 μl of IP buffer, pre-cleared with protein A-agarose, and incubated overnight at 4 °C with the following antibody: for PER1, 2 μl of anti-mPER1 rabbit antiserum (Millipore, #AB2201); for CRY1, 1 μg of affinity-purified anti-mCRY1 guinea pig polyclonal antibody[19,54]; for CRY2, 1 μg of affinity-purified anti-mCRY2 guinea pig polyclonal antibody[19,54]; and for CLOCK, 1 μg of anti-mCLOCK mouse mono-clonal antibody[55] (MBL, #D349-3). To collect immunocomplexes, 40 μl of Protein A/G Plus-agarose beads (Santa Cruz) were added and further incubated at 4 °C for 1.5 h. After beads washing and DNA elution[53], eluted DNA fragments were purified with Qiaquick Nucleotide Removal Kit (Qiagen) and subjected to qPCR analysis with *Per2* E′-box primers (forward primer, 5′-GAG GTG GCA CTC CGA CCA ATG-3′; reverse primer, 5′-GCG TCG CCC TCC GCT GTC AC-3′) and *Per2* −4.5 kb negative binding site primers (forward primer, 5′-CCA CAC GGT ACT CAG CGG GC-3′; reverse primer, 5′-GGG TCA CTG CGA GCC TTG CC-3′).

**Cell culture and immunoblotting**. WT and *Per2*E′m/m primary fibroblasts were isolated from the lung tissue of adult male mice according to a protocol established by Seluanov et al.[56]. Cells used for time course assay were between passages 4 and 6. Dispersed cells were uniformly plated in 24-well plates at a density of $1 \times 10^5$ cells per well and cultured for 2 days prior to synchronization, for which cells were treated with dexamethasone (DEX, final concentration, 200 nM) for 3 h, followed by medium refreshment at Time0. Cells were harvested every 4 h in either TRIzol reagent (Invitrogen) for RNA analysis or 2 × Laemmli buffer for Western blot analysis. Immunoblotting was performed using an affinity-purified anti-mPER2 rabbit polyclonal antibody[23] (final concentration, 2 μg ml$^{-1}$) and β-Actin antibody (A5441, Sigma, 1:1000).

**RNA extraction and RT-qPCR**. Total RNA was extracted with RNeasy kit (Qiagen) and converted to cDNA with SuperScript VILO cDNA Synthesis kit (Invitrogen). qPCR was run on a BioMark HD System (Fluidigm) with a 48.48 Fluidigm BioMark Dynamic Array chip (Fluidigm)[57]. TaqMan probe and primer sets were as follows; for *Per2*, probe, FAM-AGG CAC CTC CAA CAT GCA ACG AGC C-TAMRA, forward primer (fw): 5′-GCA CAT CTG GCA CAT CTC GG-3′, reverse primer (rv): 5′-TGG CAT CAC TGT TCT GAG TGT C-3′; for *Per2* intron, probe, FAM-TGG AGC CCA CCG CAG ACA GCC C-TAMRA, fw: 5′-CCT CTC ACC TCA TGC CCT TTT AG-3′, rv: 5′-CTG CTC AGA CCA ACA GAT TTA TCA-3′; for *Per1*, probe, FAM-AGC CCC TGG CTG CCA TGG-TAMRA, fw: 5′-CAG GCT TCG TGG ACT TGA GC-3′, rv: 5′-AGT GGT GTC GGC GAC CAG-3′; for *Per3*, probe, FAM-TTC TGC TCA TCA CCA CCC TGC GGT TCC-TA MRA, fw: 5′-ACA GCT CTA CAT CGA GTC CAT G-3′, rv: 5′-CAG TGT CTG AGA GGA AGA AAA GTC-3′; for *Cry1*, probe, FAM-TGA TCC ACA GGT CAC CAC GAG TCA GGA A-TAMRA, fw: 5′-TAG CCA GAC ACG CGG TTG-3′, rv: 5′-AGC AGT AAC TCT TCA AAG ACC TTC A-3′; for *Cry2*, probe, FAM-AGG TCT CTC ATA GTT GGC AAC CCA GGC-TAMRA, fw: 5′-TGG ACA AGC ACT TGG AAC GG-3′, rv: 5′-GGC CAG TAA GGA ATT GGC ATT C-3′; for *Clock*, probe, FAM-ACC CAG AAT CTT GGC TTT TGT CAG CAG G-C-TAMRA, fw: 5′-TGG CAT TGA AGA GTC TCT TCC TG-3′, rv: 5′-GAG ACT CAC TGT GTT GAT ACG ATT G-3′; for *Bmal1*, probe, FAM-CGC CAA AAT AGC TGT CGC CCT CTG ATC T-TAMRA, fw: 5′-GTA CGT TTC TCG ACA CGC AAT AG-3′, rv: 5′-GTA CCT AGA AGT TCC TGT GGT AGA-3′; for *Nr1d1*, probe, FAM-CCC TGG ACT CCA ATA ACA ACA CAG GTG TAMRA, fw: 5′-TCA GCT GGT GAA GAC ATG ACG-3′, rv: 5′-GAG GAG CCA CTA GAG CCA ATG-3′; for *Dbp*, probe, FAM-CGG CTC CCA GAG TGG CCC GC-TAMRA, fw: 5′-CGG CTC TTG CAG CTC CTC-3′, rv: 5′-GTG TCC CTA GAT GTC AAG CCT G-3′; for *E4bp4*, probe, FAM-CAG GGA GCA GAA CCA CGA TAA CCC ATG A-TAMRA, fw: 5′-CGC CAG CCC GGT TAC AG-3′, rv: 5′-CAT CCA TCA ATG GGT CCT TCT G-3′; for *Rplp0*, probe, FAM-TGG CAA TCC CTG ACG CAC C GC C-TAMRA, fw: 5′-GCG TCC TCG TTG GAG TGA C-3′, rv: 5′-AAG TAG T

TG GAC TTC CAG GTC G-3′. for *Gapdh*, probe, FAM-CGG CCA AAT CCG TTC ACA CCG ACC-TAMRA, fw: 5′-GAG ACG GCC GCA TCT TCT T-3′, rv: 5′-TCT CCA CTT TGC CAC TGC A-3′. Values were normalized to average expression of *Rplp0* and *Gapdh*.

**Real-time bioluminescence recording and data analysis**. Tissues from mice carrying a *Bmal1* promoter-driven luciferase reporter (*Bmal1-ELuc*)[26] were harvested from adult (2–3 months-old) male mice between ZT10–12 and cultured according to a published method[58]. In brief, brains were sliced coronally into 400-μm sections, and the paired SCN, which contains minimal volumes of the non-SCN, were cultured[59]. The lungs were cut into a small piece (~2 × 2 × 0.3 mm$^3$)[59]. The adrenal glands were sliced into a section of 0.3 mm thickness. All tissues were cultured separately on a Millicell membrane (PICMORG50, Millipore) with DMEM medium containing10 mM HEPES (pH 7.2), 2% B27 (Invitrogen) and 1 mM luciferin, in 35-mm dish, and air sealed. Bioluminescence recording was started immediately upon placement in culture with a dish-type luminometer (Kronos Dio, ATTO) maintained at 37 °C. Luminescence was measured for 3 min for each dish at 30-min intervals. The data were smoothed using a 2 h moving average and further detrended by subtracting a 24 h running average for FFT analysis. A fourth-order Blackman–Harris window was applied before the power spectrum calculation. The spectrum was normalized to an integral of one by dividing each of its elements by the sum of all elements. Circadian rhythmicity was defined as relative spectral power density at the peak in the circadian range (20–30 h). The peak values represent the power within a frequency band of 0.009.

**Locomotor activity recording and data analysis**. C57BL/6J-backcrossed *Per2*E′m/+ mice were intercrossed to produce homozygous mutant (*Per2*E′m/m) and WT (*Per2*E′+/+) progenies using in vitro fertilization. We used adult male mice (8- to 10-week old). The animals were housed individually in light-tight, ventilated closets under indicated lighting conditions with *ad libitum* access to food and water. Locomotor activity was recorded via passive infrared sensors (PIRs, FA-05F5B; Omron) with 1-min resolution and analyzed with CLOCKLAB software (Actimetrics)[60]. Free-running period in DD was determined with $\chi^2$ periodogram, based on animal behaviors in a 40-day interval taken 3 days after the start of DD condition. Rhythmicity in LL was evaluated using FFT-Relative Power (CLOCKLAB, Actimetrics). A fourth-order Blackman–Harris window was applied before the power spectrum calculation. The spectrum was normalized to an integral of one by dividing each of its elements by the sum of all elements. Circadian rhythmicity was defined as relative spectral power density at the peak in the circadian range (20–36 h). The peak values represent the power within a frequency band of 0.006. Speed of behavioral re-entrainment was evaluated using 50% phase-shift value (PS$_{50}$)[32,61]. To determine PS$_{50}$, sigmoidal dose-response curve with variable slope, $Y = \text{Bottom} + (\text{Top} - \text{Bottom})/(1 + 10^{(\log \text{PS50} - X) \text{HillSlope}})$, was fitted to the onset time points using GraphPad Prism software.

**Body temperature recording and data analysis**. Precalibrated temperature data loggers (Thermochron iButtons, DS1921H, Maxim) were surgically implanted to the peritoneal cavity of mice under general anaesthesia. After a week of recovery in LD, mice were transferred into LL. The iButtons were programmed for collecting temperature data every 20 min. For data analysis, the mean temperature of the entire data series during days 7 to 21 in LL was calculated, and the data points above the mean were displayed in double-plotted format using CLOCKLAB. For circular plot, data were detrended by subtracting a 24 h running average, and periods of top 20% temperature were deployed in Rayleigh format using Oriana 4 software (Kovacs Computer Services, UK). CT12 was extrapolated from the onsets of locomotor activity during days 3–6. Phase distribution was assessed by mean vector length (Oriana 4).

**Reporting summary**. Further information on research design is available in the Nature Research Reporting Summary linked to this article.

## Data availability

The source data underlying Figs. 1c, e, 2a, b, 3c, f, 4a–d, Supplementary Figs. 3, 4e, 6b, c, 10a, b, 11a, b are provided as a Source Data file. All the other data supporting the findings of this study are available within the article and its Supplementary Information files and from the corresponding authors upon reasonable request. A reporting summary for this article is available as a Supplementary Information file.

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

## Acknowledgements

We thank A. Bradley (Wellcome Trust Sanger Institute, UK) and Y. Nakajima (National Institute of Advanced Industrial Science and Technology, Japan) for providing ROSA26-

PBase knock-in mice and *Bmal1-Eluc* mice, respectively. This work was supported by the Core Research for Evolutional Science and Technology, Japan Science and Technology Agency (JPMJCR14W3), and the Ministry of Education, Culture, Sports, Science and Technology of Japan (15H05933, 17H01524, and 18H04015), and AMED Project for Elucidating and Controlling Mechanisms of Ageing and Longevity (JP17gm5010002, JP18gm5010002).

## Author contributions

M.D. and H.O. conceived the project and designed the research; M.D., H.S. and Y.A. contributed equally as first authors who performed experiments in collaboration with I.M., H.H., Y.T., J-M.F., Y.Y., H.K., N.K., K.Y., C.L., M.A., K.S. and H.O.; M.D. and H.O. drafted the manuscript, supported by H.S., Y.A. and J-M.F.

## Additional information

**Competing interests:** The authors declare no competing interests.

