## [Peer Review File · Nature Communications]

Reviewers' comments:

Reviewer #1 (Remarks to the Author):

In the current study, the authors mutated the E'-box within the Per2 gene and tested the effect of the mutation on the clock function in cells, the SCN and circadian behavior. The results show that a single Per2 E-box mutation affected clock function at multiple levels. Thus, the study extends previous work that focuses on protein-coding regions and demonstrates the importance of circadian regulatory cis-elements both in vitro and in vivo. The experiments are carefully designed and executed and the data are of high quality. There are no major concerns. Below are a few minor concerns and experiments suggested that would corroborate the findings.

While the main conclusion of the research is valid, the significance is slightly over stated. The requirement of the E/E'-box-mediated circadian transcription is well established (e.g. it is known that E-box mutations abolish reporter rhythms, perturbation of E-box regulators such as CLOCK and BMAL1), and the current study confirms the importance of the E-box within the Per2 gene. Other studies have shown that circadian expression of certain core clock genes/proteins is not required for the basic operation of the clock. It is a bit precarious to make general statements about the contribution of the E'-box to inducing repression; this is specific to the Per2 gene (and the altered Per2 transcriptional regulation), but not all genes that are regulated through an E-box. It is known that Per1 and Per2 transcriptional regulation is complicated and other factors contribute to their expression, not necessarily oscillatory. The title about the impact on daily maintenance of animal behavior and physiology is overstated, because again the study is limited to the Per2 E-box and limited to locomotor activity. The statement "posttranslational rhythms were also susceptible to the E'-box mutation" is not as precise as it should be; the effect is not direct because the cells are not rhythmic.

The most important finding of the study is the critical role of the Per2 E-box in generating and maintaining cell-autonomous clocks and in the SCN. Fig. 2. Fibroblasts seem completely arrhythmic, and additional data are needed to corroborate these data.

Suggest performing a Q-PCR time course analysis of peripheral tissues such as the liver and lung, as done with the fibroblasts in Fig 2B.

Suggest performing kinetic bioluminescence assays to characterize circadian rhythms in peripheral tissues such as liver and lung using WT and Per2-E'm/m;Bmal1-Eluc reporter mice, as done with the SCN in Fig 3A. These two experiments would support the fibroblast data. 3) Fig. 3A.

Suggest performing bioluminescence recording of SCN explants for a longer period of time, more than one week (medium change and record for another week or even longer), as done in Bmal1-/- SCN slices in Ko, et al, PLoS Biology, 2010, which might reveal a stronger phenotype.

Reviewer #2 (Remarks to the Author):

Doi and colleagues examine the cis-regulatory element E'box in Per2 by creating a mutant mouse line with the E'box mutated. Others have identified E boxes, non-canonical E'boxes, and the Per2 E'box described in this article as important and sufficient for driving transcriptional circadian rhythms. However, it was unclear if the Per2 E'box is necessary for circadian function. The authors demonstrate that Per2 protein and RNA are elevated in Per2 E'box mutant mice. They find that the amplitude of circadian rhythms in Bmal-Eluc reporter mice are reduced in E'box mutant mice compared to that of wild-type mice. Finally, they show that the mutant mice are less susceptible to jet-lag experiments. The data are extremely high quality and represent a tremendous effort to elegantly show that the Per2 E'box controls circadian activity through expression of Per2. The paper is suitable for publication in nature communications with only a few minor points.

There is still discrepancy about the actual transcriptional start sites of Per2:
<http://www.pnas.org/content/pnas/suppl/2005/02/04/0409763102.DC1/09763Fig6.pdf>

The UCSC genome browser lists the TSS before the Per2 E'box:
https://genome.ucsc.edu/cgi-bin/hgGene?hgsid=696764699_rDdekJW6ZBwlABn40e2rb4Nzx3UR&hgg_do_getMrnaSeq=1&hgg_gene=uc007cav.1

Please clarify why the authors choose that particular TSS.

If transcription of Per2 via the E box is abolished, then why are Per2 transcriptional levels higher? Naively, one would think that if you remove the transcriptional activator binding domain, transcription would go down, not up. Please explain.

Reviewer #3 (Remarks to the Author):

At the molecular level, the circadian clock of animals is regulated by a transcription and translation feedback loop, and this cyclic gene expression is supposed to be dependent on cis-regulatory regions of the genome. Yoo SH et al (PNAS,2005) first identified this non-canonical E-box in the mouse Per2 genome and showed that it is sufficient drive Per2 mRNA oscillations. But no one has shown that it is necessary.

To this end, Masao and colleagues used a seamless method to generate a cis-regulatory mutation of this non-canonical E-box in the mouse Per2 genome. Biochemical and behavioral experiments were carried out to systematically study the biological function of this mutation.

ChIP experiment showed CLK binding to the mutant site was drastically reduced, and qRT-PCR of nascent Per2 mRNA surprisingly implied that the function of non-canonical E-box is to induce repression. Behavioral experiments then showed this E-box is important for maintaining the rhythm stability in different light conditions. It is not however necessary for behavioral rhythmicity.

Here are some issues that should be addressed.

1. I am skeptical of the "repression" conclusion. Although not without precedent, this is not the expected result or conclusion, which would be an effect on the positive limb of the feedback loop, i.e., on transcription. I strongly suspect that the conclusion is due to the clock having stopped when the assay is being done, meaning that the effect is indirect because transcription or RNA levels can "stop" anywhere. In this regard, the authors have conveniently ignored the bottom of Fig. 2a. Why is the RNA decrease and the subsequent increase essentially normal between 16 and 28 hrs in the mutant compared to the WT? My answer: all this works well, and then the clock dies – who knows where and why.
2. I'm not a great fan of "functional compensation by Per1." If you combine two mutants both of which make the clock weak, it will be weaker even if the two proteins have different functional roles. Similarly, a weak clock can account for the poor induction of Per2 transcription, i.e., an indirect effect. However, I don't see how this can contribute to enhanced resetting.
3. Discussion: I'd say "is essential for maintaining OPTIMAL or PROPER behavioral rhythms under constant..."
4. The authors studied the mRNA and protein level of Per2 in peripheral tissues. It would be interesting to see Per2 mRNA and protein levels in the SCN.
5. The authors concluded in Fig. 2A that Per2 protein in this mutant failed to maintain normal oscillations, indeed, the electrophoretic mobility was not as good as in the control, but it looks like the overall Per2 protein levels of were still cycling. The trend would be more obvious if the film had been exposed shorter.
6. Since this mutant cannot maintain the rhythm stability in different light conditions, the authors should consider checking how Per2 responds to light alterations.

More minor comments:

7. The figure legends are not very clear, like Fig1e, which day in DD?
8. What do the error bars represent? Technical repeats or biological repeats?
9. The summary indicates "post-developmental" effects, but the mutations might also affect development and thus give rise to indirect effects. A word of caution is advised.

Re: Manuscript No: NCOMMS-18-30962-T

Non-coding *cis*-regulatory element E'-box of *Period2* is essential for daily maintenance of organismal locomotor activity and body temperature rhythmicity

Point-by-point reply:**Reviewer #1:**

In the current study, the authors mutated the E'-box within the Per2 gene and tested the effect of the mutation on the clock function in cells, the SCN and circadian behavior. The results show that a single Per2 E-box mutation affected clock function at multiple levels. Thus, the study extends previous work that focuses on protein-coding regions and demonstrates the importance of circadian regulatory cis-elements both in vitro and in vivo. The experiments are carefully designed and executed and the data are of high quality. There are no major concerns. Below are a few minor concerns and experiments suggested that would corroborate the findings.

Reply: We very much appreciate this reviewer's evaluation of our work ("*carefully designed and executed and the data are of high quality*") and his/her suggestions for further improvement. Specific concerns from this reviewer are addressed below.

1. *While the main conclusion of the research is valid, the significance is slightly overstated. The requirement of the E/E'-box-mediated circadian transcription is well established (e.g. it is known that E-box mutations abolish reporter rhythms, perturbation of E-box regulators such as CLOCK and BMAL1), and the current study confirms the importance of the E-box within the Per2 gene. Other studies have shown that circadian expression of certain core clock genes/proteins is not required for the basic operation of the clock. It is a bit precarious to make general statements about the contribution of the E'-box to inducing repression; this is specific to the Per2 gene (and the altered Per2 transcriptional regulation), but not all genes that are regulated through an E-box. It is known that Per1 and Per2 transcriptional regulation is complicated and other factors contribute to their expression, not necessarily oscillatory. The title about the impact on daily maintenance of animal behavior and physiology is overstated, because again the study is limited to the Per2 E-box and limited to locomotor activity. The statement "posttranslational rhythms were also susceptible to the E'-box mutation" is not as precise as it should be; the effect is not direct because the cells are not rhythmic.*

Reply: We thank this reviewer again. All the points are fully appreciated. Our responses to each are summarized as follows:

- We agree with this reviewer that making general statements for the contribution of the E'-box to inducing repression is not appropriate; our paper only pertains to the altered *Per2* transcriptional regulation, not describing other clock genes. To clarify this important point, we rewrote the **Summary**, **Results**, and **Discussion** as follows:

Summary [Page 2, lines 10-12] "*Without the E'-box, Per2 messenger RNA and protein expression remained at mid-to-high levels, implying that the net contribution of this particular Per2 E'-box is to reducing the expression of Per2.*"

Results [Page 7, lines 123-125] “Thus, *Per2* transcription remains active even without this particular E'-box, provoking the hypothesis that a major contribution of this particular *Per2* E'-box is to inducing circadian repression of *Per2* transcription.”.

Discussion [Page 12, lines 242-246] “Under these near-native conditions, we noticed that the deletion of the *Per2* E'-box leads to accumulation of *Per2* mRNA and protein in cultured fibroblasts. Thus, it is conceivable that the net contribution of this particular *Per2* E'-box in the circadian feedback loop is to reducing or closing the *Per2* transcription.”.

- We also concur with this reviewer that the title referring to “the impact on daily maintenance of animal behavior and physiology” is overgeneralized, because our studies are restricted to circadian measurements of locomotor activity and core body temperature. As this issue is very important – we did revise the texts not only for **Title** but also for **Summary** and **Discussion** as follows:

Title [Page 1] “Non-coding *cis*-regulatory element E'-box of *Period2* is essential for daily maintenance of organismal locomotor activity and body temperature rhythmicity”.

Summary [Page 2, lines 14-15] “... reveals the extent of the impact of the non-coding *cis*-element in daily maintenance of animal locomotor activity and body temperature rhythmicity.”.

Discussion [Page 11, lines 212-214] “... elucidate the impact of the deletion of the non-coding *cis*-element in daily maintenance of behavioral activity in adulthood.”.

- This reviewer also helped us to realize that the word “susceptible” is inappropriate to describe the effect of the E'-box mutation on “posttranslational rhythms”. We now understand this could be a secondary effect of a nonfunctional clock in the mutant cells. Taking this reviewer’s advice into account, we changed the text as follows:

Results [Page 7, lines 115-116] “The post-translational rhythms were thus affected either directly or indirectly by the mutation of the E'-box.”.

2. *The most important finding of the study is the critical role of the Per2 E-box in generating and maintaining cell-autonomous clocks and in the SCN. Fig. 2. Fibroblasts seem completely arrhythmic, and additional data are needed to corroborate these data.*

Suggest performing a Q-PCR time course analysis of peripheral tissues such as the liver and lung, as done with the fibroblasts in Fig 2B.

*Suggest performing kinetic bioluminescence assays to characterize circadian rhythms in peripheral tissues such as liver and lung using WT and *Per2-E'm/m;Bmal1-Eluc* reporter mice, as done with the SCN in Fig 3A. These two experiments would support the fibroblast data. 3) Fig. 3A.*

*Suggest performing bioluminescence recording of SCN explants for a longer period of time, more than one week (medium change and record for another week or even longer), as done in *Bmal1-/-* SCN slices in Ko, et al, PLoS Biology, 2010, which might reveal a stronger phenotype.*

Reply: Yes, we completely agree with this reviewer that additional studies with

cultured peripheral tissues would strengthen our finding. Because we had found qPCR time-course assays with separate cultured tissues (i.e. the first suggestion from this reviewer) highly variable due to inter-culture variability, we thus focused on kinetic bioluminescence reporter assays that relies on tissue cultures from *Per2E^{m/m};Bmal1-Eluc* mice (i.e. the second suggestion from this reviewer). These data are now shown in **new Fig. 3** as well as in **new Supplementary Fig. 4**. The same data are also shown below for the reviewer's perusal.

We traced luminescence from lung (**Fig. 3**) and adrenal gland (**Supplementary Fig. 4**). For comparison, original SCN data are shown together (**Fig. 3**).

■ **Figure 3 (with simplified legend)**

SCN: (a) *Bmal1-Eluc* traces of SCN cultures from *Per2E^{+/+}* ($n=5$), *Per2E^{m/m}* ($n=5$), *Per2E^{m/m}; Per1^{+/-}* ($n=5$), and *Per1^{+/-}* ($n=3$) mice. (b) FFT amplitude. (c) Daily max-to-min variations.

Lung: (d) *Bmal1-Eluc* traces of lung cultures from *Per2E^{+/+}* ($n=5$), *Per2E^{m/m}* ($n=5$), *Per2E^{m/m}; Per1^{+/-}* ($n=5$), and *Per1^{+/-}* ($n=3$) mice. (e) FFT amplitude. (f) Daily max-to-min variations.

Non-detrended raw data are also deposited in **Supplementary Fig. 4a and b**.

■ **Supplementary Figure 4 (with simplified legend)**

Adrenal: (c) *Bmal1-Eluc* traces of SCN cultures from *Per2E^{+/+}* ($n=5$), *Per2E^{m/m}* ($n=5$), *Per2E^{m/m}; Per1^{+/-}* ($n=5$), and *Per1^{+/-}* ($n=3$) mice. (d) FFT amplitude. (e) Daily max-to-min variations.

Collectively, all data obtained from lung and adrenal explant cultures are essentially equivalent to those observed for SCN and are consistent with the observations in fibroblasts. As clearly stated by this reviewer, these data constitute important additional evidence to support *the critical role of the Per2 E'-box in generating and maintaining cell-autonomous clocks*. Following the valuable suggestion from this reviewer, we clarified this point by improving the **Results** and **Discussion** as follows:

Results [Page 8, lines 151-155] “Importantly, similar attenuation of *Bmal1-Eluc* rhythmicity by *Per2E^{m/m}* and further by *Per1^{+/-}* could be reproduced in cultures of lung (**Fig. 3d-f**) and adrenal explants (**Supplementary Fig. 4**), confirming the importance of the *Per2 E'-box* in keeping sustained molecular circadian oscillations not only for the SCN but also for the peripheral tissues.”

Discussion [Page 11, lines 216-220] “*Per2 E'-box* mutation leads to unsustainable gene expression rhythms in organotypic SCN slices and cultured peripheral tissues and fibroblasts. To the best of our knowledge, these results represent the first empirical evidence that *cis*-element-based gene transcription is essential for sustaining cellular clock oscillations.”

In addition, the **Methods** of the tissue culture experiments have been included in the revised manuscript accordingly [see pages 19, lines 398-400].

- We believe that these modifications made our manuscript more comprehensive and solid in describing the role of the *Per2 E'-box* in cell-autonomous clock function.

We thought this reviewer’s third suggestion is also helpful. So we tried to establish a more refined phenotypic difference between WT and *Per2E^{m/m}* SCN slices by more long-term luminescence recording with medium change. Unfortunately, however, no additional information was obtained by such analysis.

Reviewer #2:

Doi and colleagues examine the cis-regulatory element E'box in Per2 by creating a mutant mouse line with the E'box mutated. Others have identified E boxes, non-canonical E'boxes, and the Per2 E'box described in this article as important and sufficient for driving transcriptional circadian rhythms. However, it was unclear if the Per2 E'box is necessary for circadian function. The authors demonstrate that Per2 protein and RNA are elevated in Per2 E'box mutant mice. They find that the amplitude of circadian rhythms in Bmal-Eluc reporter mice are reduced in E'box mutant mice compared to that of wild-type mice. Finally, they show that the mutant mice are less susceptible to jet-lag experiments. The data are extremely high quality and represent a tremendous effort to elegantly show that the Per2 E'box controls circadian activity through expression of Per2. The paper is suitable for publication in nature communications with only a few minor points.

Reply: We greatly appreciate this reviewer's valuable comments and positive evaluation on our work ("extremely high quality and represent a tremendous effort to elegantly show that the Per2 E'box controls circadian activity through expression of Per2"). In accordance with this reviewer's critical suggestions, we revised the paper as follows:

1. *There is still discrepancy about the actual transcriptional start sites of Per2: <http://www.pnas.org/content/pnas/suppl/2005/02/04/0409763102.DC1/09763Fig6.pdf> The UCSC genome browser lists the TSS before the Per2 E'box: https://genome.ucsc.edu/cgi-bin/hgGene?hgsid=696764699_rDdekJW6ZBwIABn40e2rb4Nzx3UR&hgg_do_getMrnaSeq=1&hgg_gene=uc007cav.1. Please clarify why the authors choose that particular TSS.*

Reply: We thank this reviewer for pointing out the definition of the TSS we chose. The TSS that we show in **Fig. 1** is exactly the same as that J.S. Takahashi group reported in 2005 (ref. 17), for which we now clarified in the revised version of our manuscript. There are several reasons why we follow this putative position originally shown by Takahashi group. Firstly, although TSS exhibits a variable nature, we did reproducibly detect this particular site as one of the main putative TSSs of *Per2* in our 5'RLM-RACE (RNA ligase-mediated rapid amplification of cDNA ends) assays with mouse liver RNA. Secondly, the same or close TSS has been used in literature (see, Proc Natl Acad Sci USA 99, 7728, 2002; Mol Biol Cell 17, 555, 2006; Nucleic Acids Res 38, 7964, 2010; Nat Commun 4, 2444, 2013). Thirdly, and equally as important as the first two, we mutagenized the CACGTT *Per2* E'-box sequence to GCTAGT, which change is also identical to that of the original paper from Takahashi group (ref. 17). Crucially, this mutation abrogates CLOCK:BMAL1-mediated promoter activation of *Per2*, in vitro (ref. 17), a rationale for choosing it for our in vivo study. For the direct comparison between in vitro and in vivo research data, we considered adoption of the same putative TSS to be helpful to avoid potential confusion of the readers of our paper.

For all intents and purposes, we respectfully prefer to continue the use of this particular TSS. Nonetheless, as pointed out clearly by this reviewer, the precise positional information of the *Per2* TSS still remains controversial. To clarify this important point, we rewrote the manuscript as follows:

Introduction [Page 4, lines 48-50] "The *Per2* E'-box sequence (5'–CACGTT–3') located near the putative transcription initiation site¹⁷ (–20 to –15) has been

demonstrated to be the principal circadian *cis*-element that is...”.

Methods [Page 14, lines 269-275] “The CACGTT E’-box sequence located in the *Per2* promoter (–20 to –15; +1, the putative transcription start site¹⁷) was targeted. Note that numbering of the relative position of the E’-box sequence will change according to the definition of the transcription initiation site^{17,18} (see also RefSeq database in the UCSC genome browser: <https://genome.ucsc.edu/>). CACGTT was mutated to GCTAGT, because this change is reported to abrogate CLOCK:BMAL1-mediated activation of *Per2* promoter activity in vitro¹⁷.”.

Methods [Page 14, lines 284-285] “+1, the putative transcription start site¹⁷.”.

Fig. 1 legend “Numbering shows the position relative to the putative transcription start site (+1).”.

2. *If transcription of Per2 via the E box is abolished, then why are Per2 transcriptional levels higher? Naively, one would think that if you remove the transcriptional activator binding domain, transcription would go down, not up. Please explain.*

Reply: We elaborate on this by adding a new paragraph to **Discussion** as follows:

[Page 12-13, lines 249-261] “*Cis*-element provides a place where both active and repressive transcription complexes are recruited. The net effect of its absence thus seems context-dependent. The question of this context specificity is still unresolved in the field of transcription and chromatin remodeling. Particularly, in our case, the mechanism(s) that allows the continued transcription of *Per2* without relying on the E’-box is still unknown. It is possible that native chromatin structure of the *Per2* might permit its transcription. It is also conceivable that other circadian *cis*-elements on the *Per2* promoter, such as D-box and CRE, might contribute to the basal transcription of *Per2*^{9,18,42}. In this regard, the continued expression of *Per2* might reflect a confounding effect of being chronically deficient in the functional E’-box. Given that *Per2* is under regulation of interlocked feedback cycles^{6,9}, remaining *Per2* transcription might be a homeostatic consequence of a nonfunctional clock in the mutant cells. A complete understanding of circadian regulation of *Per2* in vivo under native conditions remains a challenge for future study.”.

In addition to the above discussion, we now softened our argument on the role of the *Per2* E’-box by changing all relevant statements in the manuscript as follows:

Summary [Page 2, lines 10-12] “Without the E’-box, *Per2* messenger RNA and protein expression remained at mid-to-high levels, implying that the net contribution of this particular *Per2* E’-box is to reducing the expression of *Per2*.”.

Results [Page 7, lines 123-125] “Thus, *Per2* transcription remains active even without this particular E’-box, provoking the hypothesis that a major contribution of this particular *Per2* E’-box is to inducing circadian repression of *Per2* transcription.”.

Discussion [Page 12, lines 242-246] “Under these near-native conditions, we noticed that the deletion of the *Per2* E’-box leads to accumulation of *Per2* mRNA and protein in cultured fibroblasts. Thus, it is conceivable that the net contribution of this particular *Per2* E’-box in the circadian feedback loop is to

reducing or closing the *Per2* transcription.”.

We believe that these modifications have made our manuscript more precise than before in arguing the role of the *Per2* E'-box in regulating the *Per2* transcription.

Reviewer #3:

At the molecular level, the circadian clock of animals is regulated by a transcription and translation feedback loop, and this cyclic gene expression is supposed to be dependent on cis-regulatory regions of the genome. Yoo SH et al (PNAS,2005) first identified this non-canonical E-box in the mouse Per2 genome and showed that it is sufficient drive Per2 mRNA oscillations. But no one has shown that it is necessary.

To this end, Masao and colleagues used a seamless method to generate a cis-regulatory mutation of this non-canonical E-box in the mouse Per2 genome. Biochemical and behavioral experiments were carried out to systematically study the biological function of this mutation.

ChIP experiment showed CLK binding to the mutant site was drastically reduced, and qRT-PCR of nascent Per2 mRNA surprisingly implied that the function of non-canonical E-box is to induce repression. Behavioral experiments then showed this E-box is important for maintaining the rhythm stability in different light conditions. It is not however necessary for behavioral rhythmicity.

Here are some issues that should be addressed.

Reply: We thank Reviewer#3 for having had a careful reading of our manuscript. He/she provided us a lot of valuable suggestions and insightful advice, all of which helped us to improve our paper.

1. *I am skeptical of the “repression” conclusion. Although not without precedent, this is not the expected result or conclusion, which would be an effect on the positive limb of the feedback loop, i.e., on transcription. I strongly suspect that the conclusion is due to the clock having stopped when the assay is being done, meaning that the effect is indirect because transcription or RNA levels can “stop” anywhere. In this regard, the authors have conveniently ignored the bottom of Fig. 2a. Why is the RNA decrease and the subsequent increase essentially normal between 16 and 28 hrs in the mutant compared to the WT? My answer: all this works well, and then the clock dies – who knows where and why.*

Reply: Through this insightful advice from this reviewer, we now realize that our original “repression” conclusion was overstated in our previous manuscript. As noted by this reviewer, there is another important possibility: the homeostatic accumulation of *Per2* could be an indirect effect of a nonfunctional clock in the mutant cells. We have now emphasized this argument accordingly in the manuscript. Please note that, following the request of Reviewers 1 (point1) and 2 (point2), we have also extensively revised our *repression* model, by adding a new paragraph to the **Discussion** as follows:

[Page 12-13, lines 249-261] *“Cis-element provides a place where both active and repressive transcription complexes are recruited. The net effect of its absence thus seems context-dependent. The question of this context specificity is still unresolved in the field of transcription and chromatin remodeling. Particularly, in*

our case, the mechanism(s) that allows the continued transcription of *Per2* without relying on the E'-box is still unknown. It is possible that native chromatin structure of the *Per2* might permit its transcription. It is also conceivable that other circadian *cis*-elements on the *Per2* promoter, such as D-box and CRE, might contribute to the basal transcription of *Per2*^{9,18,42}. In this regard, the continued expression of *Per2* might reflect a confounding effect of being chronically deficient in the functional E'-box. Given that *Per2* is under regulation of interlocked feedback cycles^{6,9}, remaining *Per2* transcription might be a homeostatic consequence of a nonfunctional clock in the mutant cells. A complete understanding of circadian regulation of *Per2* in vivo under native conditions remains a challenge for future study.”

In addition to the above discussion, taking this reviewer’s point into account, we have softened our “*repression*” statement in all relevant places as follows:

Summary [Page 2, lines 10-12] “Without the E'-box, *Per2* messenger RNA and protein expression remained at mid-to-high levels, implying that the net contribution of this particular *Per2* E'-box is to reducing the expression of *Per2*.”

Results [Page 7, lines 123-125] “Thus, *Per2* transcription remains active even without this particular E'-box, provoking the hypothesis that a major contribution of this particular *Per2* E'-box is to inducing circadian repression of *Per2* transcription.”

Discussion [Page 12, lines 242-246] “Under these near-native conditions, we noticed that the deletion of the *Per2* E'-box leads to accumulation of *Per2* mRNA and protein in cultured fibroblasts. Thus, it is conceivable that the net contribution of this particular *Per2* E'-box in the circadian feedback loop is to reducing or closing the *Per2* transcription.”

Moreover, please note that following your later comment, we have also refined our description on the protein blot data in **Fig. 2a**. Please see our reply to Point#5.

2. *I’m not a great fan of “functional compensation by Per1.” If you combine two mutants both of which make the clock weak, it will be weaker even if the two proteins have different functional roles. Similarly, a weak clock can account for the poor induction of Per2 transcription, i.e., an indirect effect. However, I don’t see how this can contribute to enhanced resetting.*

Reply: These comments are all critical. We therefore removed the word “*functional compensation*” and use “likely dependent on” *Per1* instead. In addition, we now refer to a potential “*indirect effect*” of a weak clock to enhance the clarity of our statement as follows:

[Page 8, lines 148-151] “These remaining rhythms were likely dependent on *Per1*, because SCN explants from *Per2E^{m/m};Per1* heterozygous null mice (*Per2E^{m/m};Per1^{+/-}*; *Bmal1-Eluc*) exhibited even fewer persistent rhythms, which ...”

[Page 10, lines 195-198] “The underlying mechanism of this extremely large shift is unknown. While this could be due to an indirect effect of the weak clock, a limit cycle oscillator model³³ predicts that a reduced-amplitude pacemaker in the mutant mice (reduced radius of the limit cycle) could have this effect³⁴.”

3. Discussion: I'd say "is essential for maintaining OPTIMAL or PROPER behavioral rhythms under constant..."

Reply: Thank you for your helpful suggestion. We revised this text as follows:

[Pages 10-11, lines 208-210] "... is essential for maintaining optimal locomotor activity rhythms under constant light conditions and enabling the phase of the clock to resist abrupt shifts in LD cycling."

4. The authors studied the mRNA and protein level of *Per2* in peripheral tissues. It would be interesting to see *Per2* mRNA and protein levels in the SCN.

Reply: Addressing this interesting suggestion is somehow complex, for which we apologize.

In the manuscript, we explored the effects of the E'-box mutation on 'cell-autonomous' clock function using cultured peripheral tissues and *Bmal1-Eluc* SCN slices (please see new Fig. 3 data and our reply to Reviewer#1's point #2 request). As an important extension of this series of research, we appreciate the value of studying *Per2* expression with cultured SCN slices, which should be informative as was suggested. However, although we performed the suggested experiments, we noticed that circadian tracing of gene expression with cultured SCN slices is technically difficult, due to inter-culture/tissue variation, which unfortunately preclude providing reliable group data from separate SCN samples.

This is unlikely our specific problem. As far as we searched, there is no literature that convincingly performed circadian mRNA expression analysis with cultured SCN slices. As this reviewer is also certainly aware, luciferase reporter system recapitulating the endogenous *Per2* expression (e.g. PER2::LUC knock-in) would be useful in general, but in our case, intercrossing of the *Per2* E'-box mutation and PER2::LUC is virtually impossible because of the proximity between the two loci.

One possible alternative approach would involve studying *Per2* expression in vivo in the SCN. Yet, a concern exists that as reported in literature, in vivo phenotypes are less profound than those in vitro due to cell-nonautonomous compensatory mechanisms (refs 12, 28, 29). Indeed, for this reason, we had not expected the *Per2* E'-box deficiency to cause the abolition of in vivo *Per2* oscillation. At the organismal level, the mutant mice were able to maintain essentially unimpaired rhythmicity in LD and DD.

With that said, however, in the hope of providing potential insight for the query of the reviewer, we began to examine in vivo *Per2* mRNA expression profiles. Shown below are the data supposed for the reviewer's perusal:

This figure shows circadian expression profiles of *Per2* mRNA and pre-mRNA in the SCN on the second day in DD. Data for CT20 are double-plotted. We sampled the SCN by laser-microdissection. As was expected, *Per2* expression was basically circadian in vivo, even for the mutant mice (red). We noted that in the SCN of the mutant mice, both *Per2* mRNA and pre-mRNA show phase-advanced expression with increased baseline.

Similar observations were also observed for the liver *Per2*. The data on the left are double-plotted expression profiles of *Per1* and *Per2* in the liver in DD. *Per2* was phase-advanced with increased baseline in the mutant mice, whereas *Per1* was essentially unimpaired by the mutation.

In aggregate, our primary impression from the data above is that remaining rhythms in the *Per2* expression in the mutant mice echoes our original assumption that defective cellular clock function could not be directly evaluated at the organismal level.

Consequently, as already expected in literature, there must be compensatory mechanisms in vivo that we have not yet systematically investigated, representing an intrinsic limitation of in vivo assays in studying cell-autonomous clock function.

For the same reason, although slightly elevated or de-repressed expression of *Per2* mRNA in the mutant SCN and liver might be of potential interest, these results do not help to prove or disprove our hypothesis on the function of the E'-box.

On the basis of these considerations on the limitation of in vivo assessment, we are still conservative not to display this preliminary in vivo dataset in the current manuscript. Because we would like to leave this important issue for future study, we rewrote the paper as follows:

Discussion [Page 13, lines 259-261] “...., remaining *Per2* transcription might be a homeostatic consequence of a nonfunctional clock in the mutant cells. A complete understanding of circadian regulation of *Per2* in vivo under native conditions remains a challenge for future study.”.

■ We believe that although we could not directly address the above issue, the current manuscript that has incorporated additional studies with cultured peripheral tissues (lung and adrenal; new Fig. 3 and Supplementary Fig. 4) provide further evidence to reinforce our main conclusion that the *Per2* E'-box is essential for maintaining cell-autonomous clock function. We would also like to emphasize that possible mechanisms of in vivo compensation have been discussed in the full paragraph of **Discussion** as follows:

[Page 11, lines 221-236] “Not surprisingly, the overall behavioral phenotypes of the *Per2* E'-box mutant mice reveal that the *Per2* E'-box is not an absolute requirement for behavioral rhythm generation. Under DD conditions, its deletion affects only the circadian period (**Supplementary Figs 5 and 6**), a phenotype analogous to that observed in transgenic *Drosophila* strains expressing period or timeless via a constitutive promoter^{12,28}. These data suggest that organisms have a compensatory mechanism to mitigate defective transcriptional feedback regulation. Correspondingly, previous studies demonstrate that behavioral rhythms do not necessarily reflect cellular clock phenotypes^{29,35-37}; Superior circadian performance of behavioral rhythmicity has been observed in several clock gene mutant mice, compared to tissue cultures obtained from the same animal strains²⁹. In vivo multicellular and/or inter-organ systemic circuitry might compensate for poor core clock function within individual cells⁴. In this regard, it

is worth noting that systemic extracellular signals are known to affect the *Per2* promoter via non-E'-box *cis*-regulatory elements, such as Ca²⁺/cAMP response element (CRE)³⁸ and glucocorticoid response element (GRE)³⁹. Extracellular circadian feedback pathways through these non-E'-box elements might contribute to the compensation mechanisms *in vivo*^{40,41}.”.

5. *The authors concluded in Fig. 2A that Per2 protein in this mutant failed to maintain normal oscillations, indeed, the electrophoretic mobility was not as good as in the control, but it looks like the overall Per2 protein levels of were still cycling. The trend would be more obvious if the film had been exposed shorter.*

Reply: In agreement with this reviewer’s impression, the protein blot of the mutant cells in Fig. 2a could read an increase at 56 h. However, we note that after 60 h the mutant cells couldnt show any appreciable oscillation. To confirm this notion further, we provide western blot data of WT and mutant cells from 60 h to 92 h. These represent three independent biological replicates, with different exposure, for each genotype.

Note that WT cells appeared rhythmic even after 60 h. In comparison, PER2 proteins in the mutant cells were relatively variable or not circadian, which supports our original conclusion that the mutation of the *Per2* promoter E'-box renders the cellular clock less sustainable.

Results are now shown in **Supplementary Fig 3**. Based on these results and taking this reviewer’s important advice into account, we revised the texts in **Results** as follows:

Results [Pages 6-7, lines 109-113] “It is important to note that the mutant cells could competently form the first surge of PER2 expression after synchronization (4-12 h). However, following a subsequent increase (28 h), PER2 expression in the mutant cells remained at mid-to-high levels and eventually lost apparent circadian variation after 60 h (see also **Supplementary Fig. 3**).”.

6. *Since this mutant cannot maintain the rhythm stability in different light conditions, the authors should consider checking how Per2 responds to light alterations.*

Reply: Yes. The data are shown in **Supplementary Fig. 10** (see also its replica in the next page). We determined expression profiles of *Per2* in the SCN of WT (+/+) and

Per2^{E'm/m} (m/m) mice after a 30-min light pulse exposure at CT14. As reported, in the WT mice, *Per2* was increased about 2.5 fold 2 h after the illumination (*blue*). In comparison, in the mutant mice, *Per2* showed a greater increase of expression of approximately 4 fold (*orange*), suggesting that this increase might partly contribute to

Supplementary Figure 10

the enhanced resetting of the mutant mice. On the other hand, the mutant mice were almost equivalent to WT mice in light-induced induction of *Per1* expression in the SCN. It appears that the effect of the *Per2* E'-box mutation is specific to *Per2*. These data are now mentioned in the revised figure legend.

As mentioned earlier, while there was a slightly enhanced *Per2* mRNA induction in the mutant SCN, it is our view that this alone is unlikely to explain the extremely large phase-shift of the mutant mice. On the basis of these considerations and taking this reviewer's important comment into account, we rewrote our statement in the revised version of our manuscript as follows:

Results [Page 10, lines 195-201] "The underlying mechanism of this extremely large shift is unknown. While this could be due to an indirect effect of the weak clock, a limit cycle oscillator model³³ predicts that a reduced-amplitude pacemaker in the mutant mice (reduced radius of the limit cycle) could have this effect³⁴. We also noticed that after a light pulse exposure, the mutant mice show a slightly enhanced *Per2* mRNA induction in the SCN (see **Supplementary Fig. 10**). This perhaps also partly contributes to the enhanced resetting of the mutant mice."

Supplementary Fig. 10 legend "Note that the *Per2*^{E'm/m} mutation augments light-induced expression of *Per2* but not *Per1*."

[Continuance of our reply to point 6]

We understand that our current analysis is limited to "acute" response of *Per2* to light. Although it would be potentially informative to study *Per2* expression under chronic light exposure (LL) as well as experimental jet-lag regimes, we feel that a comprehensive set of tests encompassing different time-of-day points over multiple days or weeks (for LL and Jet-lag) would be beyond the scope of the present work. We therefore would like to address this interesting point in future studies.

More minor comments:

7. The figure legends are not very clear, like Fig1e, which day in DD?

Reply: This is the first day in DD. This information is now included in the figure legend.

8. What do the error bars represent? Technical repeats or biological repeats?

Reply: We apologize for this oversight. We now clarified that the bars represent

technical repeats. More specifically, we repeated chromatin/DNA shearing with equal amounts of aliquots from the same liver nuclear sample at each time point. In order to clarify this point, we amended the legend and rewrote the **Methods** as follows:

Methods [Page 16, lines 315-317] “ChIP assay was performed as described^{19,53} with technical repeats: We repeated chromatin/DNA shearing with equal amounts of aliquots from the same liver nuclear sample.”.

Methods [Page 16, lines 325-329] “The nuclei were resuspended in 1.5 ml per liver of IP buffer (10 mM Tris-HCl pH ...) and divided equally into three aliquots, which were each separately sonicated around 15 seconds for 80 times...”.

Fig. 1 legend “Values are means \pm s.e.m. of three technical replicates.”.

9. *The summary indicates “post-developmental” effects, but the mutations might also affect development and thus give rise to indirect effects. A word of caution is advised.*

Reply: We thank this reviewer. We removed the word “*post-developmental*” and used “in adults” instead. The text in the revised **Summary** is as follows:

[Page 2, lines 2-3] “Non-coding *cis*-regulatory elements are essential determinants of development, but their exact impacts on behavior and physiology in adults remain elusive.”.

Reviewers' comments:

Reviewer #1 (Remarks to the Author):

The authors have carefully addressed all my concerns including performing new experiments as requested.

Reviewer #2 (Remarks to the Author):

The authors have satisfactorily addressed my concerns about the location of the TSS, and their reasoning behind their choice of TSS. The data about increased Per2 mRNA and protein levels are still unresolved questions, but the authors have (correctly) softened their claims that the Per2 E'box acts as a direct repressor, and that the repression may proceed indirectly or as a result of a broken clock. Nevertheless, the data provide exceptionally high quality in vivo evidence to complement the in vitro data regarding the Per2 E'box and are suitable for publication in Nature Communications. Because the SCN mRNA data was not available, the authors should consider including the liver data they present in the response to reviewers as the second-most common tissue analyzed in the circadian rhythm field.

Reviewer #3 (Remarks to the Author):

I find the reply of the authors and the revisions quite unsatisfying. Although they do soften somewhat their wording to accommodate the criticisms of the reviewers, they really haven't changed their claims. First, they still maintain that PER is contributing to the activation of transcription. As I said in my first review, the clock has died and so who knows in what state each biochemical process will be. To be blunt, I think this continued claim is nonsense and so should not just be softened but totally excised from the paper. Second, the mutant clock is still functional as can be seen in the first or second day of constant darkness. It is just weaker and indeed damps more rapidly than the wild-type control. It is also long period, consistent with a weaker clock. The authors choose to examine their western blots quite late into DD, to emphasize the flat biochemical rhythms, but cycling is quite robust earlier in DD. In summary, these weaker rhythms and damping constitute the only substantial finding of the paper in my view.

Point by point responses to the Reviewers:

We thank our reviewers for taking their time and for their constructive criticism on our work. We have attended to the reviewers' comments and are submitting the revised manuscript with indications of where we amended (Blue, sentences added; Orange, sentences deleted). Point-by-point responses to the reviewers are provided below.

Reviewer #1:

The authors have carefully addressed all my concerns including performing new experiments as requested.

Reply: We are very glad to hear that our manuscript has been able to address all this reviewer's previous concerns.

Reviewer #2:

The authors have satisfactorily addressed my concerns about the location of the TSS, and their reasoning behind their choice of TSS. The data about increased Per2 mRNA and protein levels are still unresolved questions, but the authors have (correctly) softened their claims that the Per2 E'box acts as a direct repressor, and that the repression may proceed indirectly or as a result of a broken clock. Nevertheless, the data provide exceptionally high quality in vivo evidence to complement the in vitro data regarding the Per2 E'box and are suitable for publication in Nature Communications. Because the SCN mRNA data was not available, the authors should consider including the liver data they present in the response to reviewers as the second-most common tissue analyzed in the circadian rhythm field.

Reply: In response to this reviewer's kind suggestion, we have now included the liver mRNA data in new **Supplementary Fig. 11**. We greatly appreciate this reviewer's positive evaluation on our work ("the data provide exceptionally high quality in vivo evidence to complement the in vitro data regarding the Per2 E'box and are suitable for publication in Nature Communications").

Reviewer #3:

I find the reply of the authors and the revisions quite unsatisfying. Although they do soften somewhat their wording to accommodate the criticisms of the reviewers, they really haven't changed their claims. First, they still maintain that PER is contributing to the activation of transcription. As I said in my first review, the clock has died and so who knows in what state each biochemical process will be. To be blunt, I think this continued claim is nonsense and so should not just be softened but totally excised from the paper. Second, the mutant clock is still functional as can be seen in the first or second day of constant darkness. It is just weaker and indeed damps more rapidly than the wild-type control. It is also long period, consistent with a weaker clock. The authors choose to examine their western blots quite late into DD, to emphasize the flat biochemical rhythms, but cycling is quite robust earlier in DD. In summary, these weaker rhythms and damping constitute the only substantial finding of the paper in my view.

Reply: The points are well taken. According to this reviewer's advice, we have deleted our hypothetical claim and increased the clarity of our statement on a role for the *Per2* promoter E'-box cis-element in keeping sustainable cellular oscillation.

Please see the revised manuscript, in which we have:

- deleted the original sentence in lines 10-12 from **Summary**.
- increased the clarity in describing our substantial finding by changing **Summary** statement from "our work delineates the *Per2*-E'-box as a critical nodal element for cell-autonomous circadian rhythm generation" to "our work delineates the *Per2*-E'-box as a critical nodal element for keeping sustainable cell-autonomous circadian oscillation"
- deleted the original sentence in lines 125-127 from **Results**.
- deleted the original sentence in lines 248-250 from **Discussion**.

Please also confirm that the paragraph that we added in response to the Reviewer#3's advice remains in the current **Discussion** (see lines 253-264). This discussion paragraph refers to the interpretation of our data based on the idea that "the clock has died and so who knows in what state each biochemical process will be". We believe that thanks to this reviewer's critical comments and criticism our paper has become more solid in data presentation and discussion.

REVIEWERS' COMMENTS:

REVIEWER 1:

The authors have carefully addressed all my concerns including performing new experiments as requested.

REVIEWER 2:

The authors have satisfactorily addressed my concerns about the location of the TSS, and their reasoning behind their choice of TSS. The data about increased Per2 mRNA and protein levels are still unresolved questions, but the authors have (correctly) softened their claims that the Per2 E'box acts as a direct repressor, and that the repression may proceed indirectly or as a result of a broken clock. Nevertheless, the data provide exceptionally high quality in vivo evidence to complement the in vitro data regarding the Per2 E'box and are suitable for publication in Nature Communications. Because the SCN mRNA data was not available, the authors should consider including the liver data they present in the response to reviewers as the second-most common tissue analyzed in the circadian rhythm field.

REVIEWER 3:

I find the reply of the authors and the revisions quite unsatisfying. Although they do soften somewhat their wording to accommodate the criticisms of the reviewers, they really haven't changed their claims. First, they still maintain that PER is contributing to the activation of transcription. As I said in my first review, the clock has died and so who knows in what state each biochemical process will be. To be blunt, I think this continued claim is nonsense and so should not just be softened but totally excised from the paper. Second, the mutant clock is still functional as can be seen in the first or second day of constant darkness. It is just weaker and indeed damps more rapidly than the wild-type control. It is also long period, consistent with a weaker clock. The authors choose to examine their western blots quite late into DD, to emphasize the flat biochemical rhythms, but cycling is quite robust earlier in DD. In summary, these weaker rhythms and damping constitute the only substantial finding of the paper in my view.